# A Simple, Fast Algorithm for Continual Learning from High-Dimensional Data

**Neil Ashtekar & Vasant Honavar**
Department of Computer Science
The Pennsylvania State University
{nca5096,vuh14}@psu.edu

## Abstract

As an alternative to resource-intensive deep learning approaches to the continual learning problem, we propose a simple, fast algorithm inspired by adaptive resonance theory (ART). To cope with the curse of dimensionality and avoid catastrophic forgetting, we apply incremental principal component analysis (IPCA) to the model's previously learned weights. Experiments show that this approach approximates the performance achieved using static PCA and is competitive with continual deep learning methods. Our implementation is available on https://github.com/neil-ash/ART-IPCA.

## 1 Introduction

In the continual learning setting, we wish to efficiently learn a sequence of tasks without forgetting how to perform previously learned tasks. Many approaches have been proposed to address this problem, with recent work focused mainly on deep learning (Kirkpatrick et al., 2017; Rebuffi et al., 2017; Yoon et al., 2018). While these approaches boast impressive predictive power, they are limited by a host of drawbacks including high computational costs, large memory requirements, and poor performance on nontrivial continual learning benchmarks with long task sequences and task-identification information withheld (Farquhar & Gal, 2018; Díaz-Rodríguez et al., 2018).

We propose an alternative, matching-based algorithm which combines ideas from ART with IPCA to improve predictive performance for high-dimensional data. ART is an attractive solution to continual learning because of its proven stability guarantees which imply that catastrophic forgetting will not occur with respect to the training data (Grossberg, 2020). PCA is a well-known linear dimensionality reduction technique which can be effectively used as a data preprocessing step for both supervised and unsupervised learning tasks. Many incremental versions of PCA have been proposed to efficiently update principal component estimates without recomputing eigenvectors from the full covariance matrix (Weng et al., 2003; Zhao et al., 2006; Ross et al., 2008).

## 2 Proposed Model

We begin by formally defining our notation and problem statement. We are given a set of tasks $\mathcal{T}_1 \ldots \mathcal{T}_T$ where each task $\mathcal{T}_i$ has a corresponding dataset $D_i = (\boldsymbol{X}_i, \boldsymbol{y}_i)$. We focus on supervised classification tasks with features $\boldsymbol{X}_i \in \mathbb{R}^{n_i \times d}$ and categorical labels $\boldsymbol{y}_i \in \mathbb{N}^{n_i \times 1}$, though our proposed model can easily be extended to unsupervised clustering. We reduce our data's dimension from $d \to k$ using IPCA matrix $\boldsymbol{P} \in \mathbb{R}^{k \times d}$ whose rows consist of the top-$k$ estimated eigenvectors of the data's covariance matrix. The weights in the original space and IPCA reduced space are denoted by $\boldsymbol{w}_{org} \in \mathbb{R}^{d \times 1}$ and $\boldsymbol{w}_{red} \in \mathbb{R}^{k \times 1}$ respectively, with $\boldsymbol{w}_{red}^i = \boldsymbol{P}_i \boldsymbol{w}_{org}$ denoting the weight in reduced space under iteration $i$ of IPCA.

Our proposed model does not require task identifiers, though this information may be optionally included to improve performance. It is applicable to class-incremental, task-incremental, and domain-incremental learning, and only requires that features have consistent dimensionality across all tasks. Algorithm 1 describes the training procedure while the inference procedure and intuition behind the model are discussed in Appendix A.1. Time and space complexity are given in Appendix A.2.

---

**Algorithm 1** Supervised training in batch mode

---

1: **Input:** Set of tasks $\mathcal{T}_1 \ldots \mathcal{T}_T$, each task $\mathcal{T}_i$ has its own set of training data $D_i = (\boldsymbol{X}_i, \boldsymbol{y}_i)$, pairwise similarity function $S(\cdot, \cdot)$, vigilance $\rho$, reduced dimension $k$, learning rate $\beta$
2: $W = \emptyset$        ▷ Weights in original space
3: $C = \emptyset$        ▷ Map from weights to labels
4: **for** each task $\mathcal{T}_i$ **do**
5:      Update $\boldsymbol{P}_{i-1}$ with new data $\boldsymbol{X}_i$ to get $\boldsymbol{P}_i$
6:      **for** each training example $(\boldsymbol{x}_j, y_j) \in (\boldsymbol{X}_i, \boldsymbol{y}_i)$ **do**
7:          $\boldsymbol{w}^*_{org} = \underset{\boldsymbol{w}_{org} \in W}{\operatorname{argmax}} S(\boldsymbol{P}_i \boldsymbol{x}_j, \boldsymbol{P}_i \boldsymbol{w}_{org})$      ▷ Similarity in reduced space
8:          **if** $S(\boldsymbol{P}_i \boldsymbol{x}_j, \boldsymbol{P}_i \boldsymbol{w}^*_{org}) \geq \rho$ and $C[\boldsymbol{w}^*_{org}] = y_j$ **then**
9:             $\boldsymbol{w}^*_{org} = \beta \boldsymbol{x}_j + (1 - \beta) \boldsymbol{w}^*_{org}$      ▷ Learn weights in original space
10:          **else**
11:             $W = W \cup \boldsymbol{x}_j$      ▷ Set new weight to current sample
12:             $C[\boldsymbol{x}_j] = y_j$      ▷ Map new weight to correct label
13:          **end if**
14:      **end for**
15: **end for**

---

When applying IPCA as a preprocessing step in the continual learning setting, a key question emerges. When data from a new task arrives, the IPCA matrix is updated, resulting in a new representation of previously learned data. *How should the model's previously learned parameters be adjusted given the new representation of the training data in order to avoid catastrophic forgetting?*

We answer this question by observing that the updated version of IPCA can be applied to transform the model's weights (learned in the original space) since the weights are simply a linear combination of training samples. In other words, learning in the current IPCA reduced space results in the same weights as previously learning in the original space, then applying the current version of IPCA. This argument is formalized in Proposition 1 with proof and further explanation in Appendix A.3.

**Proposition 1.** *Let $\boldsymbol{x}_1 \ldots \boldsymbol{x}_m$ be the training samples which have matched to weight $\boldsymbol{w}$ under some previous IPCA transformation(s) $\boldsymbol{P}_1 \ldots \boldsymbol{P}_{i-1}$ [1]. Assuming that these samples $\boldsymbol{x}_1 \ldots \boldsymbol{x}_m$ still match to weight $\boldsymbol{w}$ under the current IPCA transformation $\boldsymbol{P}_i$, then learning the weight $\boldsymbol{w}^i_{red}$ in the reduced space is equivalent to first learning the weight $\boldsymbol{w}_{org}$ in the original space, then applying IPCA $\boldsymbol{P}_i$.*

## 3 EMPIRICAL RESULTS

We evaluate our proposed algorithm on the Split MNIST and Split Fashion-MNIST benchmarks in the single-headed setting with task identifiers unavailable at inference. Our proposed IPCA model is compared against two variations: (1) with no PCA and (2) with static PCA learned on the entire dataset (serving as expected lower and upper bounds for IPCA performance). The results in the top half of Table 1 are consistent with these expectations. To contextualize our model's performance, we include results from Sokar et al. (2021) on several well-known continual deep learning methods in the bottom half of Table 1. Please see Appendix A.5 for further discussion and experimental details.

Table 1: Single-headed setting – mean multiclass classification accuracy and standard deviation

| | MNIST | F-MNIST |
|---|---|---|
| No PCA | $92.99 \pm 0.41$ | $76.09 \pm 0.50$ |
| IPCA | $94.03 \pm 0.26$ | $77.56 \pm 0.60$ |
| Static PCA | $94.04 \pm 0.24$ | $77.48 \pm 0.39$ |
| EWC | $20.01 \pm 0.06$ | $19.47 \pm 0.98$ |
| iCaRL | $94.57 \pm 0.11$ | $80.70 \pm 1.29$ |
| DEN | $56.95 \pm 0.02$ | $31.51 \pm 0.04$ |

---

[1]The phrase "sample $\boldsymbol{x}$ matches to weight $\boldsymbol{w}$" means that $\boldsymbol{w}$ is the weight with the maximum similarity score with respect to $\boldsymbol{x}$ in line 7 of Algorithm 1. Weight $\boldsymbol{w}$ is also referred to as the "winner" of ART's competition.

ACKNOWLEDGEMENTS

This work was supported in part by grants from the National Science Foundation (IIS 2041759 and IIS 2226025) to Vasant Honavar and from the National Institutes of Health (NCATS UL1 TR002014).

URM STATEMENT

The first author of this work meets the URM criteria of ICLR 2023 Tiny Papers Track.

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

# A  APPENDIX

## A.1  ALGORITHM DETAILS AND INTUITION

A verbal description of Algorithm 1 is as follows. Each time a new task arrives, the IPCA matrix is updated using all of the task's data. Sequentially, for each training sample in the task, the best-matching weight is computed using a similarity metric in the IPCA reduced space of dimension $k$. If the best-matching weight is sufficiently similar to the sample and it corresponds to the correct class, then the weight is updated in the original space of dimension $d$. Otherwise, a new weight is initialized to the current sample in the original space, with its corresponding class label recorded. At inference, simply compute similarity scores between the test sample and all learned weights in the reduced space, then predict the class corresponding to the weight with the largest similarity score.

The main idea behind our proposed algorithm is to learn weights in the original space and perform matching in the IPCA reduced space. Storing the weights in the original space allows the model to adapt to changes in the input representation caused by IPCA updates. This adaptation is done by simply applying the current version of IPCA to the previously learned weights.

## A.2  TIME AND SPACE COMPLEXITY

The space complexity of our model during training is $\mathcal{O}(kd + Wd)$ where $W$ is the total number of learned weights. During training, both the current IPCA matrix and weights in the original space must be stored. The time complexity for training is $\mathcal{O}(Nkd + TWkd + NWk)$ given $N$ total training samples and $T$ tasks. This consists of the time required for learning/applying IPCA updates to the data, applying IPCA updates to the weights, and performing matching.

The space complexity of our trained model is $\mathcal{O}(kd + Wk)$. The time complexity for inference on a single sample is also $\mathcal{O}(kd + Wk)$. This is because the IPCA matrix and learned weights must be stored in order to transform new data and compute matches at inference. These bounds assume that the learned weights are stored in the reduced space of dimension $k$ after all training is complete – this is a small implementation detail included in the time complexity given above for training. Note that the number of weights $W$ can be controlled by setting the vigilance parameter $\rho$, with smaller values of $\rho$ resulting in fewer weights and coarser granularity.

## A.3  APPROXIMATION OF STATIC PCA

Proposition 1 attempts to establish an equivalence between learning incrementally in the original space and learning statically in the reduced space. If Proposition 1 holds for all weights $w \in W$ and for all iterations $i = 1 \ldots T$, then the weights learned in the IPCA reduced space will be equivalent to those learned in the static PCA reduced space, modulo the approximation error incurred by IPCA. Here, "static PCA" refers to first learning and applying PCA to all of the training data, then learning the weights solely in the PCA reduced space.

*Proof of Proposition 1.* This equivalence can be easily shown using linearity, with coefficients $\alpha$ determined by the learning rate:

$$
\begin{aligned}
\boldsymbol{w}_{red}^i &= \alpha_1 \boldsymbol{P}_i \boldsymbol{x}_1 + \ldots + \alpha_m \boldsymbol{P}_i \boldsymbol{x}_m \\
&= \boldsymbol{P}_i (\alpha_1 \boldsymbol{x}_1 + \ldots + \alpha_m \boldsymbol{x}_m) \\
&= \boldsymbol{P}_i \boldsymbol{w}_{org}
\end{aligned}
\tag{1}
$$

The top line in 1 represents learning in the current IPCA reduced space, while the bottom two lines represent previously learning in the original space, then applying the current version of IPCA. $\square$

In practice, it is unlikely that Proposition 1 will hold due to its strong assumption that matches will be the same across various iterations of IPCA. However, we do expect matches to be similar (though not exactly the same) across iterations of IPCA. Our experimental results in Table 1 demonstrate that this approximate equivalence holds in practice.

## A.4 RELATION TO ART

Our proposed algorithm is inspired by ART, and essentially performs incremental winner-take-all clustering. However, there are several differences between our algorithm and ART-based algorithms. Our algorithm differs from ART as it (1) does not include distinct bottom-up and top-down matching phases, (2) uses linear rather than fuzzy intersection weight updates, and (3) avoids match tracking. Specifically, our algorithm uses the alternative to match tracking described in Anagnostopoulos & Georgiopoulos (2003) which satisfies Fuzzy ARTMAP's incremental learning principle.

Because of these differences, some of ART's properties do not apply to our proposed algorithm. For example, the use of fuzzy intersection weight updates ensures that categories only shrink, implying that weights do not repeat previously held values (Carpenter et al., 1992). (We do not use fuzzy intersection as it is nonlinear, thus incompatible with Proposition 1). Still, our algorithm retains arguably the most important property of ART: the use of competition to protect previously learned knowledge. For further details on the ART framework and its applications in machine learning, we direct readers to this review article (Grossberg, 2020) and survey (da Silva et al., 2019).

## A.5 DETAILS AND DISCUSSION OF EMPIRICAL RESULTS

### A.5.1 PROBLEM SETTING

We evaluate our model on the commonly used Split-MNIST and Split-Fashion-MNIST datasets in Tables 1 and 2. Each dataset has 10 classes, divided into 5 binary classification problems: classify 0 vs 1, 2 vs 3, 4 vs 5, 6 vs 7, and 8 vs 9. Results are reported as the average performance across each of the 5 tasks after learning all tasks sequentially (i.e. tasks $1-5$ are first learned in order, then task 1 is evaluated, then task 2 is evaluated, etc).

Our evaluation was conducted in the single-headed setting with task-identifiers unavailable to the model. In our setting, this is equivalent to class-incremental learning (CIL). We chose this setting because it satisfies rigorous definitions of continual learning and is significantly more challenging than the multi-headed setting (Farquhar & Gal, 2018) in which task identifiers are provided at inference. Note that the multi-headed setting can be trivially solved by training isolated models for each task. For completeness, we include our proposed model's performance in the multi-headed setting, equivalent to task-incremental learning (TIL), in Table 2. Many continual deep learning approaches achieve comparably strong performance in this easier setting (Hsu et al., 2018).

Table 2: Multi-headed setting – mean binary classification accuracy and standard deviation

|  | MNIST | F-MNIST |
| --- | --- | --- |
| No PCA | $99.47 \pm 0.05$ | $98.86 \pm 0.09$ |
| IPCA | $99.46 \pm 0.07$ | $98.86 \pm 0.10$ |
| Static PCA | $99.41 \pm 0.06$ | $98.79 \pm 0.13$ |

### A.5.2 PCA VARIATIONS

In the top half of Table 1 and in Table 2, we compare our proposed IPCA model against variations with no form of PCA and with static PCA trained on the entire dataset. Assuming that PCA improves matching performance, we expect the "No PCA" and "Static PCA" models to serve as approximate lower and upper bounds for "IPCA" model performance. In the single-headed setting, we find that our proposed model with IPCA significantly outperforms the model learned without PCA and does not significantly differ from the model learned with static PCA (using paired $t$-tests with significance level $\alpha = 0.05$). In the easier multi-headed setting, we find that all models offer strong performance.

### A.5.3 DEEP LEARNING BASELINES

To understand our proposed model's performance in the context of recent work on continual learning, we include three deep learning baselines in the bottom half of Table 1. We include one method from each of the three main categories of continual learning approaches. Elastic weight consolidation (EWC) (Kirkpatrick et al., 2017) is a regularization-based approach, incremental classifier and representation learning (iCaRL) (Rebuffi et al., 2017) is a rehearsal-based approach, and dynamically expandable networks (DEN) (Yoon et al., 2018) is an architecture-based approach. These results are adopted from Sokar et al. (2021), which includes results from Hsu et al. (2018) and van de Ven & Tolias (2019). We include these baselines to contextualize our model's performance. While we do not expect our model to outperform all deep learning methods in terms of predictive accuracy, our results show that we achieve comparable performance with a much simpler approach.

### A.5.4 EXPERIMENTAL DETAILS

The following describes how we evaluated our proposed algorithm in Tables 1 and 2. We use 2000 train samples and 500 test samples of each class. Our results are computed over 25 trials. In each trial, the training data is randomly shuffled, resulting in different presentation orderings across trials – this is the only source of randomness in our algorithm. We use PCA/IPCA to reduce to dimensions $k = 200$ for the MNIST dataset and $k = 250$ for the Fashion-MNIST dataset, which corresponds to preserving slightly more than $95\%$ of the variance in the training data. Both datasets have original dimension $d = 784$. We use vigilance parameter $\rho = 0.5$, cosine similarity function $S(\cdot, \cdot)$, and adaptive learning rate $1/n_s$ where $n_s$ denotes the total number of samples which have matched with the winning weight (i.e. the weights are simply an average of their corresponding samples). These values were determined through basic hyperparameter tuning on a validation set with 500 samples per class. We use the scikit-learn (Pedregosa et al., 2011) implementation of IPCA based on Ross et al. (2008).

### A.6 FUTURE WORK

There are several potential extensions of our proposed algorithm. First, note that the main idea behind Algorithm 1 in Proposition 1 could conceivably be applied to other types of incremental learning algorithms beyond IPCA and ART. For example, an incremental version of linear discriminant analysis (LDA) (Pang et al., 2005) could be used as an alternative to IPCA.

Second, it would be interesting to consider hierarchical extensions of IPCA. This is a way to perform local dimensionality reduction (Chakrabarti & Mehrotra, 2000) to increase the amount of retained variance at lower dimensions. Specifically, a global version of IPCA could be applied when clustering data with ART in an unsupervised manner, then local versions of IPCA could be applied to the data within each cluster. This process can be repeated to achieve hierarchical representation learning or hierarchical clustering. Proposition 1 allows this to be done in the continual learning setting as data arrives sequentially through time.

Finally, it would be interesting to further explore methods like ART for continual learning. Recent work on continual learning is focused on retrofitting deep learning methods in order to avoid catastrophic forgetting. ART is able to avoid catastrophic forgetting due to the inherent properties of its architectures and algorithm. A deeper understanding of these properties could lead to innovative variants and extensions of ART for continual learning.

