# OpenReview forum: "A Simple, Fast Algorithm for Continual Learning from High-Dimensional Data"
_ICLR.cc/2023/TinyPapers — Submitted to Tiny Papers @ ICLR 2023_

### Official Review · Reviewer_7Yc5 · 2023-03-27

**Confidence:** 4

**Summary Of Contributions:**

This paper aims to solve the problem of Catastrophic forgetting using Adaptive Resonance Theory and Incremental PCA. The results are benchmarked on Split MNIST and Split Fashion MNIST. According to the results shared in the paper their method seems to perform on par with the Static PCA.

**Rating:**

High Potential (HP): a submission which meets the reviewing criteria and has potential to make an impact on the field

**Strengths And Weaknesses:**

### Strengths :
-  I think the paper was very well written especially the motivation behind their method was clearly explained as well as the experimental details. The notation seems to be consistent which is excellent.
- The proofs of the proposition, and the time complexity of the algorithm is certainly nice to have and adds more credibility to the paper.
-  The inclusion of algorithm and its simplicity suggests it should be easy to replicate the results, although I have not tried to do that.

### Weakness :
- I think there is a lack of discussion section/future works section that might be helpful to the readers of the paper to develop better intuition in what more could be done. It would certainly make the paper a more exciting read in my opinion.
-  There should have been at least 1 more continual method to benchmark against so a reader can fully understand the extent of the method.


**Suggested Changes:**

-  As mentioned above, having one more baseline to benchmark against would be very nice.
-  I would certainly like it if there was a future directions/discussion section to see what the authors further plan to go with their method.

Overall I do think this paper is very well written and fun to read, so super job on that to authors !

---

### Official Review · Reviewer_rQup · 2023-04-04

**Confidence:** 3

**Summary Of Contributions:**

The paper uses Incremental PCA as a methodology to apply to the Continual Learning task.

**Rating:**

High Impact (HI): a submission which meets the reviewing criteria and is predicted to make an impact on the field

**Strengths And Weaknesses:**


Pretty solid paper. Theoretical results, time and space complexity analysis, and good details in the appendix buttress the paper. It almost feels like a paper in itself.
Explain more about the ART framework.

**Suggested Changes:**


Explain more about the ART framework.

---

### Author Response · Authors · 2023-05-09
**Changes in Revised Version**

We thank the reviewers for their helpful comments and patience as we complete revisions. The following describes our main changes in the revised version:
1. Deep continual learning baselines: As suggested by reviewer 7Yc5 and area chair CxXa, we have added deep learning baseline results to contextualize our model’s performance. Specifically, we have included results on elastic weight consolidation (EWC), incremental classifier and representation learning (iCaRL), and dynamically expandable networks (DEN), all in the single-headed setting on the MNIST and Fashion MNIST datasets. These results are given in the bottom half of Table 1 and explained in greater detail in Appendix A.5.
2. Future work: As suggested by reviewer 7Yc5, we have added a section outlining potential future work in Appendix A.6.
3. Further explanation of ART: Reviewer rQup suggested we explain more about the ART framework. While we feel that an in-depth discussion of the framework is outside the scope of this tiny paper, we direct interested readers to a review article on ART (Grossberg, 2020) as well as a survey on applications of ART to machine learning (da Silva et al., 2019) in Appendix A.4.
4. Code: We include a link to our github implementation in the abstract.

---

### Author Response · Authors · 2023-05-30
**Opt-in for Archival**

We wish to to opt-in for archival of this work.

---

### Meta-Review · Area_Chair_CxXa · 2023-04-10

**Recommendation:** Invite to present
**Confidence:** 4

**Metareview:**

The paper is well-written, compact, appears to be original, and proposes a method that benefits from simplicity and fast run-times. Both reviewers agree that the paper is clear, correct, and reproducible. Reviewer 7Yc5 points out that a non-ART baseline would be important to increase a reader's clarity on how to contextualize the results table, which I agree would improve the paper.

**Summary:**

The paper proposes a fast continual learning method combines incremental PCA for feature learning and and ART-like classifier on top. The paper is well-written and contains relevant details for replication, although it is difficult to contextualize the results (all of them are from ART-like methods).

**Reason For Not Giving A Higher Recommendation:**

The paper is a nice contribution and meets CCR, with a slight revision that would add an additional baseline result (as suggested by reviewer 7Yc5), which is important to contextualize the results (which seem promising). As is, it is hard to understand what the numbers mean (unless e.g. one is already familiar with the range of continual learning performance on fashion-mnist).

I imagine there would be a result that could be added from an existing continual learning paper that tackled split-mnist / split-fashion-mnist? The concern is less whether the proposed method performs the best, but to be able to understand how it relates to the performance of e.g. some representative deep learning approach. In particular, numbers from other method applied to the more difficult single-headed version of split-fashion-mnist would be desirable (because that is the most difficult variant of the task). If there is a reason that it would be difficult to include such a baseline, an alternative would be at least to highlight how much headroom there is in these tasks (e.g. what is the general range of performance in single-headed Fashion-MNIST)?



**Reason For Not Giving A Lower Recommendation:**

N/A

---

### Decision · Program_Chairs · 2023-04-10

Invite to present